# Psychometric properties of the Malay version of the Behavioural Regulation in Exercise Questionnaire (BREQ-3)

**Shirlie Chai[1,2], Yee Cheng Kueh[3]\*, Najib Majdi Yaacob[3], Garry Kuan[4]**

**1** Pharmacy Department, Miri Hospital, Ministry of Health Malaysia, Sarawak, Malaysia, **2** Clinical Research Centre Miri, Miri Hospital, Ministry of Health Malaysia, Sarawak, Malaysia, **3** Biostatistics and Research Methodology Unit, School of Medical Sciences, Universiti Sains Malaysia, Kelantan, Malaysia, **4** Exercise and Sports Science, School of Health Sciences, Universiti Sains Malaysia, Kelantan, Malaysia

\* yckueh@usm.my

## Abstract

### Background

The present study aimed at validating the Malay-language version of the Behavioural Regulation in Exercise Questionnaire (BREQ-3M) using confirmatory factor analysis (CFA).

### Methods

Data were collected from undergraduate students in the Health Campus, Universiti Sains Malaysia. A total of 674 students completed the BREQ-3M (male: 19.4%, female: 80.6%), with a mean age of 20.27 years (SD = 1.35). Behavioural regulation was assessed with the 24-item BREQ-3M. Standard forward-backward translation was performed to translate the English version of BREQ-3 into the Malay version.

### Results

The initial measurement models tested did not result in a good fit for the data. Subsequent examination of the CFA results suggested some modifications, including adding correlations between the item residuals within the same subscale and deletion of identified regulation. These modifications resulted in good fit indices (Root Mean Square Error of Approximation, RMSEA = 0.049; Comparative Fit Index, CFI = 0.949; Tucker-Lewis Index, TLI = 0.938; Standardised Root Mean Square Residual, SRMR = 0.049). The final measurement model comprised 20 items and had significant factor loadings of more than 0.50, ranging from 0.580 to 0.868. The composite reliability ranged between 0.746–0.841 for the five-factor model.

### Conclusions

The 20-item translated version of BREQ-3M is valid and reliable for assessing the behavioural regulation for exercise among university students in Malaysia.

**Data Availability Statement:** All relevant data are within the paper and its Supporting Information files.

**Funding:** This research was supported by the Ministry of Higher Education Malaysia for Fundamental Research Grant Scheme (FRGS) with Project Code: FRGS/1/2020/SKK06/USM/03/1.

**Competing interests:** The authors have declared that no competing interests exist.

## Perspective

This study examined the psychometric properties of the Malay-language BREQ-3. It was the first to assess the measurement model in Malaysia using CFA.

## Introduction

Regular physical activity is vital in maintaining a healthy lifestyle. Insufficient physical activity is viewed as one of the most important risk factors for mortality worldwide [1,2]. World Health Organization (WHO) reported that the risk of mortality associated with non-communicable diseases is 20% to 30% higher in individuals who are insufficiently active, compared to those sufficiently active [1]. However, the occurrence of chronic diseases associated with physical activity, such as hypertension and type-2 diabetes mellitus among adolescents and young adults has increased tremendously in many parts of the world [3–6].

Understanding motivation for exercise could be useful in improving the level of physical activity. Motivation is a major correlate and potential determinant of health behaviours such as physical activity [7–13]. There are several types of motivation or behavioural regulation for the behaviours. It was initially conceived along a dimension of low to high motivation level. Deci and Ryan argued that the regulation of intentional behaviours varies along a continuum from autonomous (i.e., self-determined, to promote choice) to controlled (i.e., to pressure one toward specific outcomes) [14]. When a behaviour is autonomously motivated, the person will have a sense of volition, feeling of concurring with and an entire willingness to engage in the behaviour. The behaviour is said to be congruent with respect to the person's sense of self. On the contrary, for the behaviour which is controlled, the person would feel externally or internally pressured, forced, or compelled to act.

The Self Determination Theory (SDT) is a motivational theory which proposes a classification of three motivational types (amotivation, extrinsic motivation, and intrinsic motivation) and the associated behavioural regulations (external, introjected, identified, integrated, and intrinsic regulation) [15,16]. The concept of extrinsic and intrinsic motivation explains motivation with respect to both their inner and outer worlds. On the other hand, amotivation, which reflects a lack of intention to engage in a behaviour, is a completely non-self-determined form of self-regulation.

Ryan and Deci postulated that having some intrinsic motivation to be among the most fundamental element in sustaining exercise [17]. A meta-analysis of studies involving children and adolescents confirmed that the types of behavioural regulation is linked to participation in physical activity [18]. Autonomous forms of motivation which include intrinsic and identified regulation, have moderate, positive associations with physical activity. Meanwhile, both amotivation and the controlled forms of motivation, which comprise of introjected and external regulation, have weak, negative associations with physical activity [18].

Many researchers use the Behavioural Regulation in Exercise Questionnaire (BREQ) and its' subsequent modifications to assess persons' motivation towards exercise behaviour [19–23]. The responses are congruent with the "why" in goals pursuit [15] and the scale is valid and reliable in measuring motivational regulation in physical activity [16,24,25]. BREQ-3 has been translated into several languages, including Portuguese, Spanish, Mexican Spanish, Dutch, and Arabic [19,20,26–29]. It is, however, not available in the Malay language.

Thus, there is a need to develop a valid measurement scale to assess motivation towards the adoption of physical activity among university students, who are generally young adults. In

this study, we aimed to translate the BREQ-3 into Malay for use in the Malaysian population and to confirm the validity and reliability of the Malay-language version of BREQ-3 (BREQ-3M) among Malaysian university students.

## Methods

### Participants

We distributed a total of 715 self-administered questionnaires to the undergraduate students in the Health Campus, Universiti Sains Malaysia (USM), and 674 completed and returned the questionnaires. There were 131 males (19.4%) and 543 females (80.6%) with a mean age of 20.27 years (SD = 1.35). They identified themselves as Malay (78.3%), Chinese (14.0%), Indian (3.0%), and others (4.7%) but all were Malaysians and able to understand the Malay language. The median duration of physical activity was 90 minutes per week. Most of the participants reported the absence of comorbidities (91.0%) and were non-smokers (94.7%).

### Questionnaire translation

The original English version of the BREQ-3 was translated into Malay using the following steps. First, the third author forward translated the English version into Malay and aimed to retain the contents' meaning rather than render literal, word-to-word translation. Second, a local bilingual Malay who was competent in both languages back-translated the Malay version into English. Third, a panel of three experienced experts in sport sciences and sports psychology reviewed and finalised both versions. The panel members were competent bilingual speakers of both languages. The panel reviewed and related each item to its corresponding item in the English version. All the differences were properly addressed.

The final version of the BREQ-3M was pre-tested in ten undergraduate students to assess the clarity and questionnaire presentation. The results of the pre-test indicated no necessary modification.

### Data collection

This cross-sectional study obtained ethical approval from the USM Human Research Ethics Committee (USM/JEPeM/16080258) and was conducted in accordance with the Declaration of Helsinki. The study was carried out from December 2017 to April 2018 at the Health Campus, USM using the self-administered BREQ-3M and convenience sampling method.

During the recruitment, the researchers approached lecturers to ask for permission for questionnaire distribution at the end of the classes. Students who were interested and willing to participate remained in the classroom and completed the questionnaire. Implied consent was obtained when the participants returned the questionnaire to the researchers. The estimated time to complete the BREQ-3M was 15 minutes. The researchers collected 674 complete responses.

### Measures

**Demographic and physical activity information.** The questionnaire included items on the participants' demographic characteristics (e.g., gender, age, and ethnicity), duration of physical activity per week, presence of comorbidities, and smoking status.

**Behavioural Regulation in Exercise Questionnaire-3 (BREQ-3).** The BREQ-3 comprises 24 items to assess an individual's motivation towards exercise [16,24]. There are six forms of motivation from the SDT: amotivation, external regulation, introjected regulation, identified regulation, integrated regulation, and intrinsic motivation. Each subscale contains four items

to measure the behavioural regulatory styles. A five-point response scale, ranging from 0 (not true for me) to 4 (very true to me) applied for the item scoring.

In Spain, it had been reported to have adequate validity and reliability (30). The authors reported the fit indices to be $\chi 2$ (215, N = 524) = 689.13, $I$ = 0.00; $\chi 2$ / df = 3.20; comparative fit index = 0.91; incremental fit index = 0.91; root mean square error of approximation = 0.06; standardised root mean square residual = 0.06. Cronbach alpha ranged from 0.66 to 0.87. However, in the Portuguese version, conducted among schizophrenia patients, amotivation subscale was deleted and items were classified as controlled and autonomous motivation (two factors) [30]. Vancampfort and colleagues performed reliability testing for the BREQ-3 and obtained Cronbach alpha ranging from 0.66 for amotivation to 0.75 for integrated regulation [22]. However, in this study, the Malay version of the BREQ-3 was examined with a six-factor model based on the original English version.

Traditionally, the scale could be interpreted as a unidimensional score of the degree of self-determination, by using the relative autonomy index (RAI). The unidimensional score was based on the SDT postulate that different types of regulation and motivation are on a continuum of self-determination. The index was attained by multiplying each subscale score by its weighting according to the position on the continuum, and subsequently, summing the weighted scores [31,32].

More recently, some researchers argued that the continuum is weak, motivation types should not form a "continuous whole" (continuum) and the construct of motivation is multi-dimensional [33]. Therefore, the scale could be used as a multidimensional instrument to give discrete scores for each of the subscales. In this study, the means of the six subscales represent the level of each motivation type [31].

## Statistical analysis

Mplus version 8 was employed to analyse the confirmatory factor analysis (CFA) results. The data were pre-screened and cleaned to detect errors. We would exclude the responses with missing values of more than 5% from the analysis [34]. The final data analysis included 674 completed questionnaires. Mardia multivariate skewness and kurtosis test indicated a violation of the multivariate normality assumption. The maximum likelihood with robust standard errors (MLR) estimator selected because it is robust to the non-normality of observations [35].

The initial hypothesised measurement models with 24 observed variables (BREQ-3M items) were adopted and examined using CFA. In this study, the goodness-of-fit indices used included comparative fit index (CFI), Tucker-Lewis index (TLI), standardised root mean square residual (SRMR) and root mean square error of approximation (RMSEA) with a confidence interval (CI) of 90%. Hair et al. suggested the following cut-offs based on the sample size and the model complexity: CFI and TLI > 0.92, SRMR ≤ 0.08 (with CFI > 0.92), and RMSEA ≤ 0.07 (with CFI ≥ 0.92) (37). We referred to the CFA modification indices for model re-specification to obtain the best-fit measurement models with adequate theoretical support.

After identifying the best-fit measurement models, we assessed the construct validity. In CFA, construct validity comprised convergent validity and discriminant validity. Convergent validity was evaluated based on the factor loading of at least 0.50 and statistically significant [36]. Subscales' convergent validity was tested using composite reliability (CR) and average variance extracted (AVE). The recommended values were at least 0.70 for CR [36] and 0.50 for AVE [37]. The correlation coefficient between factors of 0.85 or less indicated acceptable discriminant validity [38].

## Results

Table 1 summarises the distribution of items of BREQ-3M. The hypothesised measurement model for the six-factor BREQ-3M consisted of 24 items. The results for the initial

**Table 1. Distribution of items in BREQ-3M.**

| Subscales | Items | Min-Max | Mean (SD) | n (%) | | | | | |
|---|---|---|---|---|---|---|---|---|---|
| | | | | 0 | 1 | 2 | 3 | 4 | Missing |
| **Amotivation** | B2. I don't see why I should have to exercise | 0–4 | .99 (1.05) | 265 (39.3) | 241 (35.8) | 89 (13.2) | 62 (9.2) | 16 (2.4) | 1 (0.1) |
| | B8. I can't see why I should bother exercising | 0–4 | 1.22 (1.18) | 225 (33.4) | 224 (33.2) | 111 (16.5) | 78 (11.6) | 35 (5.2) | 1 (0.1) |
| | B14. I don't see the point in exercising | 0–4 | 0.97 (1.09) | 290 (43.0) | 214 (31.8) | 79 (11.7) | 71 (10.5) | 16 (2.4) | 4 (0.6) |
| | B20. I think exercising is a waste of time | 0–4 | 0.94 (1.12) | 308 (45.7) | 209 (31.0) | 69 (10.2) | 62 (9.2) | 24 (3.6) | 2 (0.3) |
| **External Regulation** | B6. I exercise because other people say I should | 0–4 | 1.68 (1.16) | 114 (16.9) | 212 (31.5) | 170 (25.2) | 133 (19.7) | 45 (6.7) | 0 (0.0) |
| | B12. I take part in exercise because my friends/family/partner say I should | 0–4 | 1.84 (1.22) | 107 (15.9) | 174 (25.8) | 182 (27.0) | 144 (21.4) | 67 (9.9) | 0 (0.0) |
| | B18. I exercise because others will not be pleased with me if I don't | 0–4 | 1.21 (1.16) | 226 (33.5) | 225 (33.4) | 105 (15.6) | 93 (13.8) | 25 (3.7) | 0 (0.0) |
| | B24. I feel under pressure from my friends/family to exercise | 0–4 | 1.24 (1.21) | 233 (34.6) | 208 (30.9) | 102 (15.1) | 95 (14.1) | 35 (5.2) | 1 (0.1) |
| **Introjected Regulation** | B4. I feel guilty when I don't exercise | 0–4 | 2.12 (1.15) | 51 (7.6) | 161 (23.9) | 212 (31.5) | 153 (22.7) | 96 (14.2) | 1 (0.1) |
| | B10. I feel ashamed when I miss an exercise session | 0–4 | 1.58 (1.16) | 128 (19.0) | 214 (31.8) | 197 (29.2) | 83 (12.3) | 52 (7.7) | 0 (0.0) |
| | B16. I feel like a failure when I haven't exercised in a while | 0–4 | 1.93 (1.18) | 92 (13.6) | 147 (21.8) | 213 (31.6) | 153 (22.7) | 67 (9.9) | 2 (0.3) |
| | B22. I would feel bad about myself if I was not making time to exercise | 0–4 | 1.96 (1.14) | 77 (11.4) | 162 (24.0) | 207 (30.7) | 167 (24.8) | 60 (8.9) | 1 (0.1) |
| **Identified Regulation** | B1. It's important to me to exercise regularly | 0–4 | 3.12 (0.85) | 3 (0.4) | 18 (2.7) | 136 (20.2) | 257 (38.1) | 259 (38.4) | 1 (0.1) |
| | B7. I value the benefits of exercise | 0–4 | 3.11 (0.87) | 7 (1.0) | 24 (3.6) | 108 (16.0) | 286 (42.4) | 248 (36.8) | 1 (0.1) |
| | B13. I think it is important to make the effort to exercise regularly | 0–4 | 3.00 (0.95) | 12 (1.8) | 30 (4.5) | 143 (21.2) | 248 (36.8) | 240 (35.6) | 1 (0.1) |
| | B19. I get restless if I don't exercise regularly | 0–4 | 1.90 (1.16) | 82 (12.2) | 179 (26.6) | 197 (29.2) | 152 (22.6) | 62 (9.2) | 2 (0.3) |
| **Integrated Regulation** | B5. I exercise because it is consistent with my life goals | 0–4 | 2.47 (1.00) | 20 (3.0) | 91 (13.5) | 217 (32.2) | 242 (35.9) | 103 (15.3) | 1 (0.1) |
| | B11. I consider exercise part of my identity | 0–4 | 2.21 (1.11) | 45 (6.7) | 136 (20.2) | 212 (31.5) | 191 (28.3) | 88 (13.1) | 2 (.3) |
| | B17. I consider exercise a fundamental part of who I am | 0–4 | 2.36 (1.08) | 37 (5.5) | 107 (15.9) | 204 (30.3) | 228 (33.8) | 98 (14.5) | 0 (0.0) |
| | B23. I consider exercise consistent with my values | 0–4 | 2.45 (1.00) | 23 (3.4) | 88 (13.1) | 220 (32.6) | 245 (36.4) | 97 (14.4) | 1 (0.1) |
| **Intrinsic Regulation** | B3. I exercise because it's fun | 0–4 | 2.96 (0.93) | 7 (1.0) | 38 (5.6) | 151 (22.4) | 258 (38.3) | 219 (32.5) | 1 (0.1) |
| | B9. I enjoy my exercise sessions | 0–4 | 2.93 (0.91) | 6 (0.9) | 35 (5.2) | 164 (24.3) | 262 (38.9) | 207 (30.7) | 0 (0.0) |
| | B15. I find exercise a pleasurable activity | 0–4 | 3.01 (0.90) | 8 (1.2) | 28 (4.2) | 137 (20.3) | 276 (40.9) | 224 (33.2) | 1 (0.1) |
| | B21. I get pleasure and satisfaction from participating in exercise | 0–4 | 2.90 (0.94) | 13 (1.9) | 36 (5.3) | 148 (22.0) | 286 (42.4) | 189 (28.0) | 2 (0.3) |

**Table 2. Summary for six-factor model fit indices.**

| Path model | CFI | TLI | SRMR | RMSEA (90% CI) | Probability RMSEA |
|---|---|---|---|---|---|
| 6-Factor Model 1 | 0.881 | 0.862 | 0.080 | 0.068 (0.063, 0.072) | < 0.001 |
| 6-Factor Model 2[a] | 0.915 | 0.899 | 0.078 | 0.058 (0.053, 0.062) | 0.003 |
| 5-Factor Model[b] | 0.949 | 0.938 | 0.052 | 0.049 (0.043, 0.055) | 0.614 |

Note: CFI = comparative fit index; TLI = Tucker-Lewis index; SRMR = standardised root mean square residual; RMSEA = root mean square error of approximation.

[a] Six-Factor measurement model with correlated items residual; B6 and B12, B1 and B13, B2 and B8, B7 and B13, B1 and B7.

[b] Five-Factor measurement model with identified regulation subscale removed.

measurement model displayed a poor fit to the data (Table 2). The loadings of all items were higher than 0.50, ranging from 0.561 to 0.857, and statistically significant (*P*<0.001) (Fig 1). The initial six-factor model improved by correlating the items' residuals within the same subscale (Fig 2). The model displayed a marginally adequate fit to the data with poor standardised factor loadings, ranging from 0.381 to 0.868. There were three identified regulation indicators which had poor factor loadings. However, all of them were statistically significant (*P*<0.001). The final model (five-factor model) was established by deleting identified regulation subscale after adding the correlations on the items' residuals. In the final model, the standardised factor loading ranged between 0.580 and 0.868. The results of the five-factor model showed a good fit to the data (Fig 3).

## Convergent and discriminant validity

In the six-factor model 2, the CR ranged between 0.449 and 0.841. Identified regulation had the lowest reliability (CR = 0.449), whereas the values for the other subscales were above 0.7. The finding was consistent with the three unsatisfactory factor loadings from this subscale. Identified regulation had a poor AVE (AVE = 0.225), whereas the values for other subscales ranged between 0.480 and 0.574. Although the AVE value for external regulation was slightly less than the recommended value of 0.50, all CR values were more than 0.7. The final five-factor model, therefore, was considered to achieve convergent validity [37].

Based on the standardised covariances between factors, all factor correlations were below 0.85. Thus, discriminant validity was evident for the subscales in the five-factor model. Tables 3 and 4 present the CR, AVE values, and the correlation coefficients for the six-factor model 2 and the final five-factor model.

## Discussion

In the current study, we translated the 24-item, English version of the BREQ-3 into Malay and confirmed the questionnaire's psychometric properties among the Malay-speaking population. The tool is used to determine the behavioural regulations, which are the underlying reasons which influence the decisions to engage in physical exercise in individuals. Thus, the BREQ-3M items should accurately reflect the behavioural regulations construct.

SDT is a macro theory of human motivation and has been a mainstay within the motivational literature for more than 40 years and remains actively researched these days. SDT can be adapted to any discipline and its applications are wide, including in the field of sports and exercise [39]. Ryan and Deci posited that different forms and phenomenal sources of motivation had varied effects on the experiences and behavioural consequences or outcomes [17]. The pioneers pointed out that behavioural regulation is the "why" of goal pursuit. It captures the reason why an individual pursues his or her goal and is the motivational resource

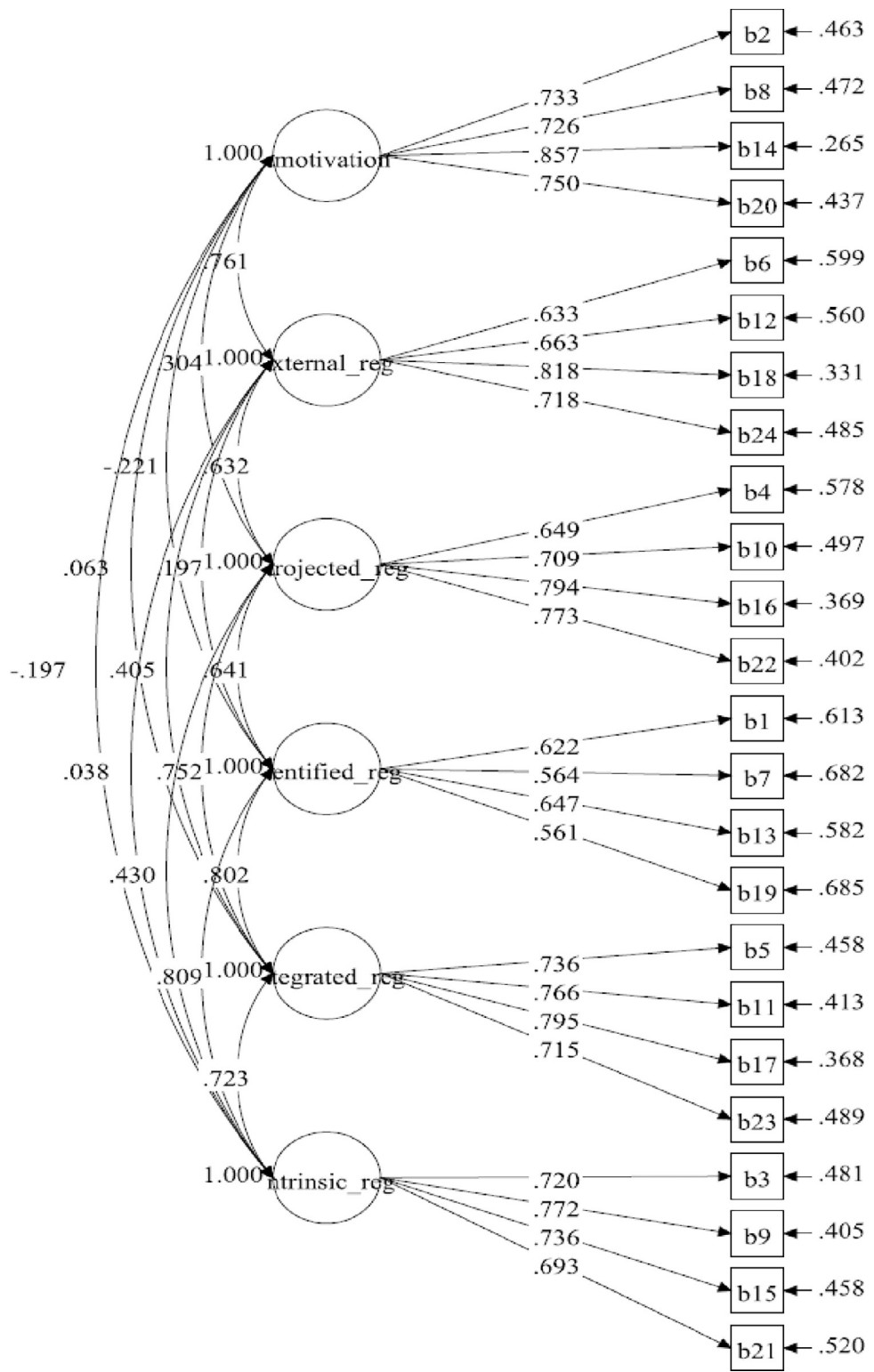

**Fig 1. BREQ-3M measurement model (six-factor model 1).**

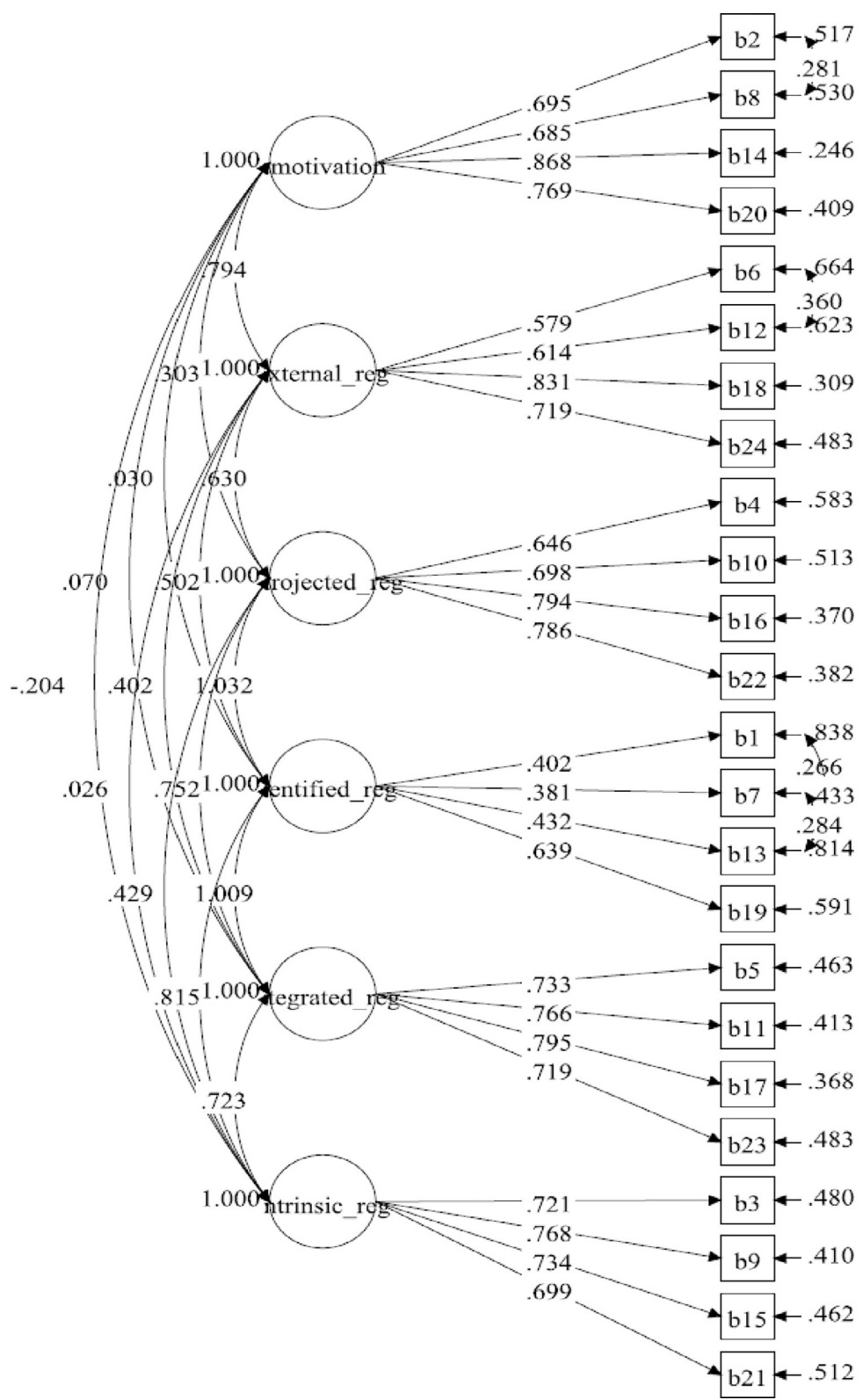

**Fig 2. BREQ-3M measurement model (six-factor model 2).**

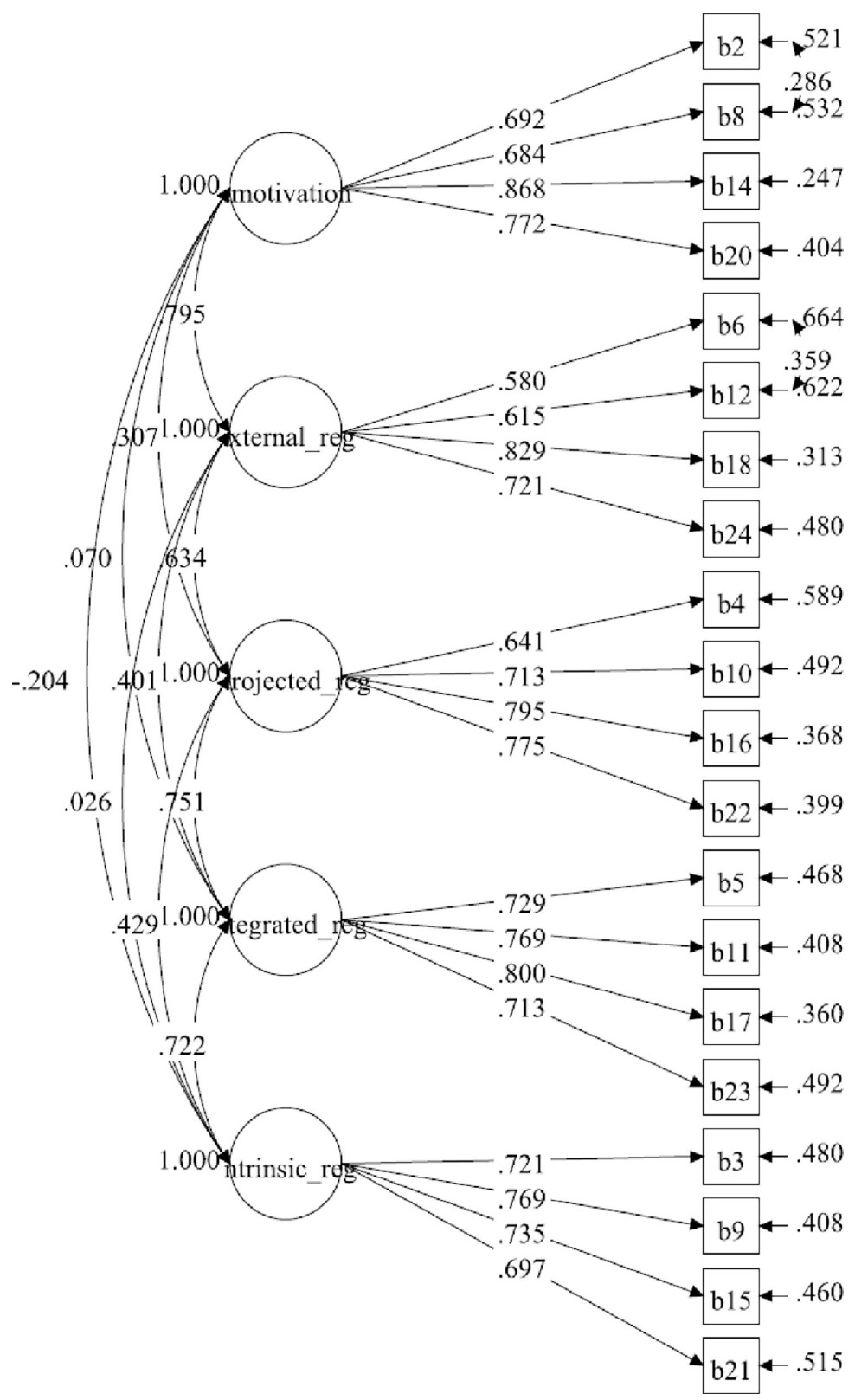

**Fig 3. BREQ-3M measurement model (five-factor model).**

**Table 3. CR, AVE, and standardised factor covariance for six-factor model 2 of BREQ-3M.**

| Subscale | CR | AVE | 1 | 2 | 3 | 4 | 5 | 6 |
|---|---|---|---|---|---|---|---|---|
| 1. Amotivation** | 0.819 | 0.574 | 1 | 0.794* | 0.303* | 0.030 | 0.070 | -0.204* |
| 2. External Regulation** | 0.746 | 0.480 | | 1 | 0.630* | 0.502* | 0.402* | 0.026 |
| 3. Introjected Regulation | 0.823 | 0.538 | | | 1 | 1.032* | 0.752* | 0.429* |
| 4. Identified Regulation** | 0.449 | 0.225 | | | | 1 | 1.009* | 0.815* |
| 5. Integrated Regulation | 0.841 | 0.568 | | | | | 1 | 0.723* |
| 6. Intrinsic Regulation | 0.820 | 0.534 | | | | | | 1 |

Note: CR = composite reliability; AVE = average variance extracted.

* Correlation is significant at the 0.05 level (two-tailed).

** Scale with error covariance.

underpinning a behaviour [15]. BREQ was first developed in 1990s [25]. It was subsequently revised by several researchers to form a six-factor scale [16,24].

Researchers had translated BREQ-3 into several languages and tested the tool in various populations, namely Portuguese, Spanish, Mexican Spanish, Dutch, and Arabic [19,20,26–29]. CFA results revealed that the initial six-factor model tested in this study required modifications. Despite correlating the error terms for items with similar meanings within the same construct, the revised six-factor model did not sufficiently fit the data. The factor loadings, CR, and AVE for identified regulation were poor. Besides, the standardised covariances with other factors, i.e. introjected regulation and integrated regulation were also higher than the recommended value.

The three items with item loading less than 0.50 in identified regulation were revisited and found to be in line with the original English version. Elimination of the three items with low loading led to a one-item construct, which is undesirable according to the three-item rule [36]. Thus, we subsequently removed the identified regulation. Model fit indices indicated a good fit to the data. In addition, the factor loadings for the five-factor model performed relatively better than the six-factor model, were above 0.50, and were statistically significant.

The AVE, which represented the mean of the squared item loadings for each subscale, ranged between 0.480 and 0.574 for the five-factor model. External regulation had an AVE value lower than the recommended 0.50. However, the construct was considered to have adequate convergent validity as the CR was 0.746, greater than the level of 0.70 suggested by Hair et. al [36]. The correlation values of the five-factor model were all less than the recommended value of 0.85, suggesting good discriminant ability [38].

**Table 4. CR, AVE, and standardised factor covariance for the final five-factor model of BREQ-3M.**

| | CR | AVE | 1 | 2 | 3 | 5 | 6 |
|---|---|---|---|---|---|---|---|
| 1. Amotivation** | 0.819 | 0.574 | 1 | 0.795* | 0.307* | 0.070 | -0.204* |
| 2. External Regulation** | 0.746 | 0.480 | | 1 | 0.634* | 0.401* | 0.026 |
| 3. Introjected Regulation | 0.823 | 0.538 | | | 1 | 0.751* | 0.429* |
| 5. Integrated Regulation | 0.841 | 0.568 | | | | 1 | 0.722* |
| 6. Intrinsic Regulation | 0.820 | 0.534 | | | | | 1 |

Note: CR = composite reliability; AVE = average variance extracted.

* Correlation is significant at the 0.05 level (two-tailed).

** Scale with error covariance.

Findings in the translational studies of the previous BREQ varied. In the original BREQ-2 study, the item "I get restless if I don't exercise regularly" in identified regulation was omitted due to an unspecified error [24,40]. However, the item was also inconsistent in Spanish, Portuguese, and Greek studies. It was ultimately removed due to poor loading factor [41,42], poor model fit, high standard error for parameter estimate, and standardised residual [40]. Nonetheless, González-Cutre and Sicilia argued that it should be measuring introjection regulation, thus placing the item in introjected regulation in their Spanish version [29]. However, the item performed well on identified regulation in the six-factor model 2, with the standardised path coefficient of 0.639.

In the subsequent version of BREQ-2R, the identified regulation items loaded satisfactorily onto their intended subscale with the value of 0.60 to 0.87 (study 1) and 0.57 to 0.83 (study 2) while Cronbach's alpha coefficient for identified regulation was 0.78 (study 1) and 0.70 (study 2). Nonetheless, identified regulation had the lowest reliability coefficients compared to other subscales in both studies [16].

We observed that the other three items from identified regulation were inconsistent in the sample. However, our findings showed that the item added into BREQ-3, item "I consider exercise consistent with my values" in the integrated regulation, was consistent with the study by Costa and colleagues [30]. However, amotivation was removed due to cross-loadings of items. The final exploratory factor analysis revealed a two-factor model, representing autonomous and controlled regulations.

There are two major limitations to this study. First, the data were collected from a single university, which might limit the generalisability of the results to other Malaysian populations. Validation of the finding in a different population to assess the result consistency to confirm the applicability of the model is worth considering in future studies. A similar finding may indicate that identified regulation might be theoretically derived but not culturally appropriate in the local context. Second, the self-report questionnaires may bring disadvantages such as potential response bias and social desirability bias. However, we attempted to avoid such biases by the reassurance of the anonymity and confidentiality of the information.

The present study confirmed that the 20-item, five-factor BREQ-3M is valid and reliable to be used to evaluate the motivational resources underpinning exercise behaviour. Nevertheless, it is vital to further examine the replicability of BREQ-3M in various Malay-speaking populations.

## Conclusion

The final measurement model for the BREQ-3M tested had shown to be a valuable measurement tool for the different forms of motivational regulations. Identified regulation subscale removed. 20 items and five subscales were retained and confirmed to be fit for the sample data. Researchers, exercise educators, and sports psychologists may use the BREQ-3M to evaluate levels of the behavioural regulations in exercise among people whose main spoken language is Malay.

## Supporting information

**S1 BREQ data. Data.**
(PDF)

## Author Contributions

**Conceptualization:** Shirlie Chai, Yee Cheng Kueh, Garry Kuan.

**Formal analysis:** Shirlie Chai, Yee Cheng Kueh.

**Investigation:** Shirlie Chai, Yee Cheng Kueh, Garry Kuan.

**Methodology:** Shirlie Chai, Yee Cheng Kueh, Najib Majdi Yaacob, Garry Kuan.

**Project administration:** Shirlie Chai, Garry Kuan.

**Resources:** Garry Kuan.

**Software:** Yee Cheng Kueh.

**Supervision:** Yee Cheng Kueh, Najib Majdi Yaacob, Garry Kuan.

**Writing – original draft:** Shirlie Chai.

**Writing – review & editing:** Shirlie Chai, Yee Cheng Kueh, Najib Majdi Yaacob, Garry Kuan.

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
