## [Decision Letter · Decision Letter 0]

24 Nov 2021

PONE-D-21-23771

Psychometric Properties of the Malay Version of the Behavioral Regulation in Exercise Questionnaire (BREQ-3)

PLOS ONE

Dear Dr. Kueh,

Thank you for submitting your manuscript to PLOS ONE. After careful consideration, we feel that it has merit but does not fully meet PLOS ONE’s publication criteria as it currently stands. Therefore, we invite you to submit a revised version of the manuscript that addresses the points raised during the review process.

We look forward to receiving your revised manuscript.

Kind regards,

Mohammad Asghari Jafarabadi

Academic Editor

PLOS ONE

Reviewers' comments:

Reviewer's Responses to Questions

**Comments to the Author**

1. Is the manuscript technically sound, and do the data support the conclusions?

Reviewer #1: Yes

Reviewer #2: Yes

2. Has the statistical analysis been performed appropriately and rigorously? 

Reviewer #1: Yes

Reviewer #2: Yes

3. Have the authors made all data underlying the findings in their manuscript fully available?

Reviewer #1: Yes

Reviewer #2: Yes

4. Is the manuscript presented in an intelligible fashion and written in standard English?

Reviewer #1: Yes

Reviewer #2: No

5. Review Comments to the Author

Reviewer #1: I would like to mention the following comments:

1-Abstract: RMSEA and SRMR must be defined.

2- Introduction: "Insufficient physical

activity is viewed as one of the most important risk factors for mortality worldwide" needs a reference.

3- Introduction: The third paragraph: "Study showed" which study?

4- Introduction is relatively long.

5- Methods: "There are 131 males" It might be better to change all tenses to "past".

6- USM needs a definition.

7- "RMSEA" and "V" should be defined in the first place. They must also be defined at the bottom of table 2.

8- "AVE" and "CR" must be defined in the text and at the bottom of tables 3 and 4 (not in the title).

9- There is no explanation about various Bs (B2, B6, ...). Specially in table 1.

10- "The composite reliability ranged between 0.746 – 0.841 for the five-factor model." How about the other models? If all 5 factors would not available, what will be the next step?

Good Luck

Reviewer #2: - Statistical analyses were done rigorously and appropriately.

- Discussions were made clear and results were properly explained.

- Writing style could be revised for better readability and consistency. Past tense, present tense and future tense were inconsistently used throughout the paper and within each paragraph.

- Authors reported having received 674 responses; yet, also reported to have excluded responses (without reporting the number of removal) and leading to the final 674 data analyzed. This brings to an incongruency with the collection and removal of data reported, or confusion to the readers.

6. PLOS authors have the option to publish the peer review history of their article (what does this mean?). If published, this will include your full peer review and any attached files.

Reviewer #1: **Yes: **Masoud Amiri

Reviewer #2: No

---

## [Author Response · Author response to Decision Letter 0]

22 Jan 2022

Dear Mohammad Asghari Jafarabadi,

On behalf of my co-authors, I would like to thank you for your consideration of our manuscript, titled “Psychometric Properties of the Malay Version of the Behavioral Regulation in Exercise Questionnaire (BREQ-3)”. We thank the Editor and the Reviewers for the careful consideration of our manuscript and for the largely favourable appraisal of this work. We also appreciate the Editor’s and Reviewers’ comments and think that the comments substantially helped us enhance the paper’s overall quality. As such, we highly appreciate the opportunity to revise and resubmit the paper. We hope that we have now adequately addressed the Editor’s and reviewer’ concerns and that you will now find our manuscript a useful addition to PLoS One. However, should there be any further issues requiring our attention, we would be grateful for a further opportunity to rectify these. Below, you will find our detailed responses to the comments. 

Editor’s comments and authors’ responses:

Journal staff:

Response: We have ensured that the manuscript meets PLOS ONE’s style requirement. 

Response: We have added one reference (reference 2) and updated the reference list in the revised manuscript. 

Reviewer #1:

1-Abstract: RMSEA and SRMR must be defined.

Response: 

Thanks to the reviewer for pointing out our mistake. The full terms for the abbreviations have been spelled out in the abstract.

2- Introduction: "Insufficient physical activity is viewed as one of the most important risk factors for mortality worldwide" needs a reference.

Response:

References were added for the statement.

3- Introduction: The third paragraph: "Study showed" which study?

Response:

We deemed the sentence as unnecessary and hence removed the sentence to improve the clarity.

4- Introduction is relatively long.

Response: We have trimmed the introduction from 642 words to 536 words.

5- Methods: "There are 131 males" It might be better to change all tenses to "past".

Response:

We thank the reviewer for the comment. We have corrected the tenses accordingly.

6- USM needs a definition.

Response:

The definition of USM is available in Method section, Participants subsection, line 106.

7- "RMSEA" and "V" should be defined in the first place. They must also be defined at the bottom of table 2.

Response:

We have added the full terms for the abbreviations in the Method section, line 151-152. We have also added the definition below Table 2. Thanks for pointing out the omission.

8- "AVE" and "CR" must be defined in the text and at the bottom of tables 3 and 4 (not in the title).

Response:

The full terms for the abbreviations were spelled out in the Method section, line 193. We have also added the definition below Tables 3 and 4. Thanks for pointing out the omission.

9- There is no explanation about various Bs (B2, B6, ...). Specially in table 1.

Response:

Thanks to the reviewer. We have added the item explanation in Table 1.

10- "The composite reliability ranged between 0.746 – 0.841 for the five-factor model." How about the other models? If all 5 factors would not available, what will be the next step?

Response:

In this study, we calculated the composite reliability for the tested models reported in the manuscript. The composite reliability for six-factor model was mentioned in line 220 in the following sentence.

“In the six-factor model 2, the CR ranges between 0.449 and 0.841”. The composite reliability is also available in Table 2.

CR for six-factor model 1 is not needed as the model fitness has not been satisfactorily achieved.

We are not exactly understand what reviewer meant by “If all 5 factors would not available, what will be the next step?”. But we will try to answer in here. If the 5 factors is not available or not fit to the data in this study, we will discuss and get further opinion from the experts and/or contacted the original author of the questionnaire. We will not proceed with further item delection or removing subscale. 

Reviewer #2: - Statistical analyses were done rigorously and appropriately.

- Discussions were made clear and results were properly explained.

- Writing style could be revised for better readability and consistency. Past tense, present tense and future tense were inconsistently used throughout the paper and within each paragraph.

Response:

We thank the reviewer for the comment. We have revised the use of tenses accordingly.

- Authors reported having received 674 responses; yet, also reported to have excluded responses (without reporting the number of removal) and leading to the final 674 data analyzed. This brings to an incongruency with the collection and removal of data reported, or confusion to the readers.

Response:

The details were added in the line 105-106 in the Method section.

---

## [Decision Letter · Decision Letter 1]

21 Feb 2022

PONE-D-21-23771R1Psychometric Properties of the Malay Version of the Behavioral Regulation in Exercise Questionnaire (BREQ-3)PLOS ONE

Dear Dr. Kueh,

Thank you for submitting your manuscript to PLOS ONE. After careful consideration, we feel that it has merit but does not fully meet PLOS ONE’s publication criteria as it currently stands. Therefore, we invite you to submit a revised version of the manuscript that addresses the points raised during the review process.

We look forward to receiving your revised manuscript.

Kind regards,

Mohammad Asghari Jafarabadi

Academic Editor

PLOS ONE

Journal Requirements:

Reviewers' comments:

Reviewer's Responses to Questions

**Comments to the Author**

1. If the authors have adequately addressed your comments raised in a previous round of review and you feel that this manuscript is now acceptable for publication, you may indicate that here to bypass the “Comments to the Author” section, enter your conflict of interest statement in the “Confidential to Editor” section, and submit your "Accept" recommendation.

Reviewer #1: All comments have been addressed

Reviewer #2: (No Response)

2. Is the manuscript technically sound, and do the data support the conclusions?

Reviewer #1: Yes

Reviewer #2: Yes

3. Has the statistical analysis been performed appropriately and rigorously? 

Reviewer #1: Yes

Reviewer #2: Yes

4. Have the authors made all data underlying the findings in their manuscript fully available?

Reviewer #1: Yes

Reviewer #2: Yes

5. Is the manuscript presented in an intelligible fashion and written in standard English?

Reviewer #1: Yes

Reviewer #2: No

6. Review Comments to the Author

Reviewer #1: I think the authors have addressed all comments in their response to reviewers. They have considered all my comments one by one.

Reviewer #2: Line 109-110: Recommend to mention the measures used to determine "sound reading and speaking skills in Malay", be it self-reported or any other methods. If none were used, please omit this line.

Line 141-142: Would prefer a more congruent reporting given that you've reported information about "non-smokers" under the participants section.

Overall: The use of tenses had been corrected accordingly. However, there are still some minor grammatical errors throughout the paper. Would be good to have it revised.

7. PLOS authors have the option to publish the peer review history of their article (what does this mean?). If published, this will include your full peer review and any attached files.

Reviewer #1: **Yes: **Masoud Amiri

Reviewer #2: No

---

## [Author Response · Author response to Decision Letter 1]

13 Mar 2022

Reviewer #2: Line 109-110: Recommend to mention the measures used to determine "sound reading and speaking skills in Malay", be it self-reported or any other methods. If none were used, please omit this line.

Response:

In this study, we did not use the measure or method to determine the reading and speaking skills. However, Malay Language is compulsory subject for all students in Malaysia in school and to pass the subject to enroll in public university. Therefore, all respondents had reasonable level of reading and speaking skills in Malay. Nevertheless, to avoid confusion and improve the clarity of the manuscript, we changed the line to “able to understand Malay language”.

Line 141-142: Would prefer a more congruent reporting given that you've reported information about "non-smokers" under the participants section.

Response: 

Thank you for the suggestion. We have added “smoking status” in the measures section.

Overall: The use of tenses had been corrected accordingly. However, there are still some minor grammatical errors throughout the paper. Would be good to have it revised.

Response:

We thank the reviewer for the comment. We have revised the use of tenses accordingly.

---

## [Decision Letter · Decision Letter 2]

16 May 2022

Psychometric Properties of the Malay Version of the Behavioral Regulation in Exercise Questionnaire (BREQ-3)

PONE-D-21-23771R2

Dear Dr. Kueh,

We’re pleased to inform you that your manuscript has been judged scientifically suitable for publication and will be formally accepted for publication once it meets all outstanding technical requirements.

Kind regards,

Mohammad Asghari Jafarabadi

Academic Editor

PLOS ONE

Reviewers' comments:

Reviewer's Responses to Questions

**Comments to the Author**

1. If the authors have adequately addressed your comments raised in a previous round of review and you feel that this manuscript is now acceptable for publication, you may indicate that here to bypass the “Comments to the Author” section, enter your conflict of interest statement in the “Confidential to Editor” section, and submit your "Accept" recommendation.

Reviewer #1: All comments have been addressed

Reviewer #2: All comments have been addressed

2. Is the manuscript technically sound, and do the data support the conclusions?

Reviewer #1: Yes

Reviewer #2: Yes

3. Has the statistical analysis been performed appropriately and rigorously? 

Reviewer #1: Yes

Reviewer #2: Yes

4. Have the authors made all data underlying the findings in their manuscript fully available?

Reviewer #1: Yes

Reviewer #2: Yes

5. Is the manuscript presented in an intelligible fashion and written in standard English?

Reviewer #1: Yes

Reviewer #2: Yes

6. Review Comments to the Author

Reviewer #1: All my comments have been addressed by authors. They answered the coments one by one with the details.

Reviewer #2: Previous comments were all addressed accordingly. The manuscript has been written in a clear and concise manner with sufficient information provided.

7. PLOS authors have the option to publish the peer review history of their article (what does this mean?). If published, this will include your full peer review and any attached files.

Reviewer #1: **Yes: **Masoud Amiri

Reviewer #2: No

---

## [Editor Report · Acceptance letter]

15 Jun 2022

PONE-D-21-23771R2 

Psychometric Properties of the Malay Version of the Behavioural Regulation in Exercise Questionnaire (BREQ-3) 

Dear Dr. Kueh:

I'm pleased to inform you that your manuscript has been deemed suitable for publication in PLOS ONE. Congratulations! Your manuscript is now with our production department. 

Kind regards, 

on behalf of

Professor Mohammad Asghari Jafarabadi 

Academic Editor

PLOS ONE